# Microbiological Study of Yamal Lakes: A Key to Understanding the Evolution of Gas Emission Craters

**Alexander Savvichev [1,\*], Marina Leibman [2,3], Vitaly Kadnikov [1], Anna Kallistova [1], Nikolai Pimenov [1], Nikolai Ravin [1], Yury Dvornikov [2]**  **and Artem Khomutov [2,3]**

[1]  Winogradsky Institute of Microbiology and Institute of Bioengineering, Research Centre of Biotechnology of the Russian Academy of Sciences, Moscow 119071, Russia; vkadnikov@bk.ru (V.K.); kallistoanna@mail.ru (A.K.); npimenov@mail.ru (N.P.); nravin@biengi.ac.ru (N.R.)
[2]  Earth Cryosphere Institute Tyumen Scientific Centre SB RAS, Tyumen 625000, Russia; moleibman@mail.ru (M.L.); ydvornikow@gmail.com (Y.D.); akhomutov@gmail.com (A.K.)
[3]  Tyumen State University, Tyumen 625003, Russia
\*  Correspondence: savvichev@mail.ru; Tel.: +7-909-975-6370

**Abstract:** Although gas emission craters (GECs) are actively investigated, the question of which landforms result from GECs remains open. The evolution of GECs includes the filling of deep hollows with atmospheric precipitation and deposits from their retreating walls, so that the final stage of gas emission crater (GEC) lake development does not differ from that of any other lakes. Microbial activity and diversity may be indicators that make it possible to distinguish GEC lakes from other exogenous lakes. This work aimed at a comparison of the activity and diversity of microbial communities in young GEC lakes and mature background lakes of Central Yamal by using a radiotracer analysis and high-throughput sequencing of the 16S rRNA genes. The radiotracer analysis revealed slow-flowing microbial processes as expected for the cold climate of the study area. GEC lakes differed from background ones by slow rates of anaerobic processes (methanogenesis, sulfate reduction) as well as by a low abundance and diversity of methanogens. Other methane cycle micro-organisms (aerobic and anaerobic methanotrophs) were similar in all studied lakes and represented by *Methylobacter* and ANME 2d; the rates of methane oxidation were also similar. *Actinobacteria*, *Bacteroidetes*, *Betaproteobacteria*, and *Acidobacteria* were predominant in both lake types. Thus, GEC lakes may be identified by their scarce methanogenic population.

**Keywords:** continuous permafrost; gas emission crater; dissolved methane; microbial processes; carbon and sulfur cycles; microbial diversity; high-throughput sequencing of the 16S rRNA genes

## 1. Introduction

Gas emission craters (GECs) are a specific feature of, and recently discovered permafrost phenomenon found in, West Siberia's Yamal and Gydan peninsulas. No other permafrost areas have been reported to have such features. However, some publications have suggested that GECs are similar to the sea floor structures known as pockmarks [1,2]. In recent years, GECs have been actively investigated by using such approaches as field monitoring and geological studies [2–6], remote-sensing data analysis [1,7–10], and laboratory testing and modeling [11–14]. GECs have never been described before, so one of the problems to be solved is looking for the indicators that may point to the lakes or other landforms that have resulted from GEC formation in the past. Cryogenesis can be one of the reasons for GECs' formation, as it affects methane distribution in permafrost [15]. It is well-established that the evolution of GECs, at least during the first few years after their appearance, turns them into lakes. Initially, deep hollows are filled with deposits from their retreating walls, so that, finally, crater-originating lakes

would not differ from any other lakes. One of the possible indicators of a lake's origin is a high content of dissolved methane in the lake water [2,3]. Methane released due to the thawing of permafrost, as well as the gas produced in situ in the bottom sediments by methanogenic archaea, is released into the lake water. Depending on the origin of the methane, its isotopic composition ($\delta^{13}$C-CH$_4$) differs [16]. Microbial oxidation of methane (both aerobic and anaerobic) within the water column and sediments results in the preferential consumption of methane with isotopically lighter carbon [17], thus enriching suspended organic matter with the lighter $^{13}$C isotope. Residual methane is enriched by the heavier C isotope. As a result, the isotope composition of methane and organic matter are indicators of the geochemical consequences of microbial processes in the methane cycle [18].

Although the methane concentration in a crater lake's water is initially much higher than in tested background lakes, it is decreasing with time [2], probably because the source of methane in the layer that the crater walls have exposed, which is responsible for GEC formation, freezes through, and may soon decrease to the level that is characteristic of thermokarst lakes. One of the possible methods to establish the indicators of the gas emission origin of a lake, differentiating it from thermokarst or other exogenous lakes, is the activity and diversity of microbial communities. A pioneer study was undertaken to determine the applicability of such an approach to distinguish lakes of either thermokarst or GEC origin.

Our hypothesis is that microbial activity and diversity, especially those associated with the methane cycle, would be different in old background lakes compared to new GEC lakes. Thus, a comparative microbiological study of the lakes will provide a microbial indicator of the lake's origin.

## 2. Study Area

Sampling was undertaken in two GEC lakes and two background lakes of the Yamal Peninsula, West Siberia, Russia in April 2016. GEC-1 (formed in October 2013) and GEC-2 (formed in October 2012) (Figure 1a,b) were almost entirely filled in with water and sediments 2–3 years later (Figure 1c,d).

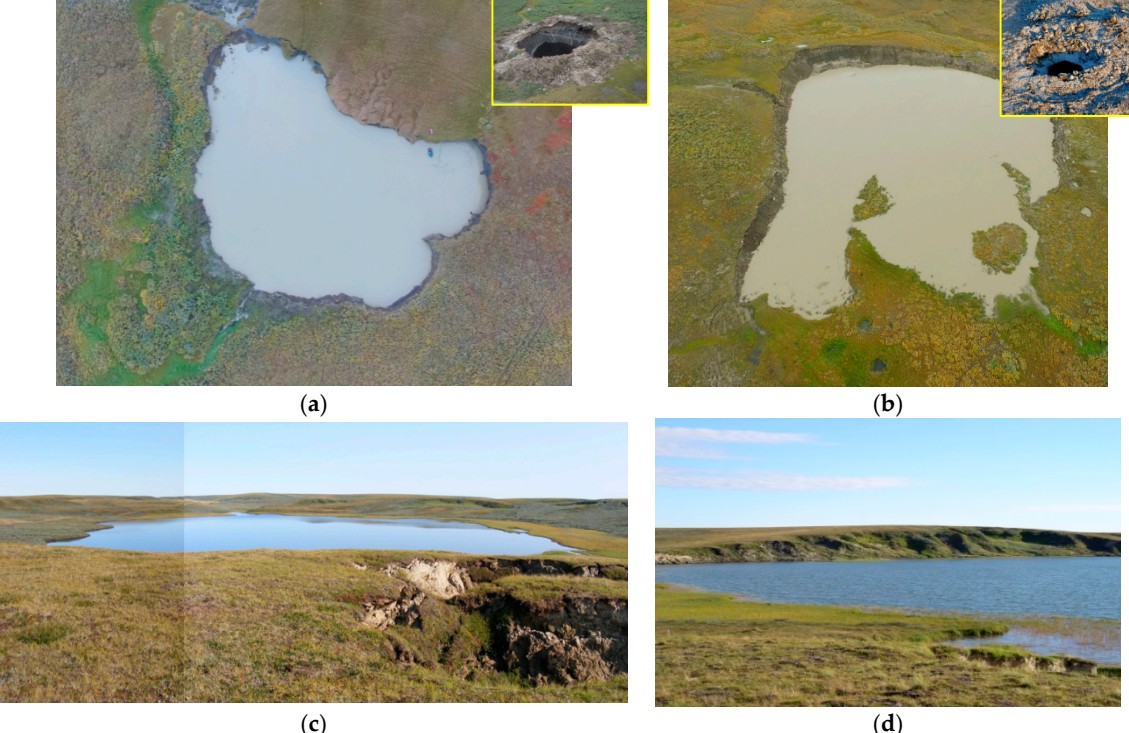

(**a**)  (**b**)

(**c**)  (**d**)

**Figure 1.** Gas emission crater (GEC) lakes and background lakes in Central Yamal: (**a**) the lake GEC-1 (the photo in the insert shows the initial state of GEC-1); (**b**) the lake GEC-2 (the photo in the insert shows the initial state of GEC-2); (**c**) background Lake 001; (**d**) background Lake 015.

The background lakes varied in their coastal erosion and, thus, sediment features. Lake 001 was larger than lake 015, its coasts were rather stable, and there was much less sediment supply compared to lake 015, the coasts of which were retreating actively due to thermodenudation. The crater lakes differed in age and, thus, in the size achieved by the time of sampling: GEC-1 was smaller and 1 year younger than GEC-2. During the process of coastal retreat, the crater walls showed rather similar, though variously deformed, clayey deposits with limited silty layers and thick layers of tabular ground ice in both craters (Figure 2).

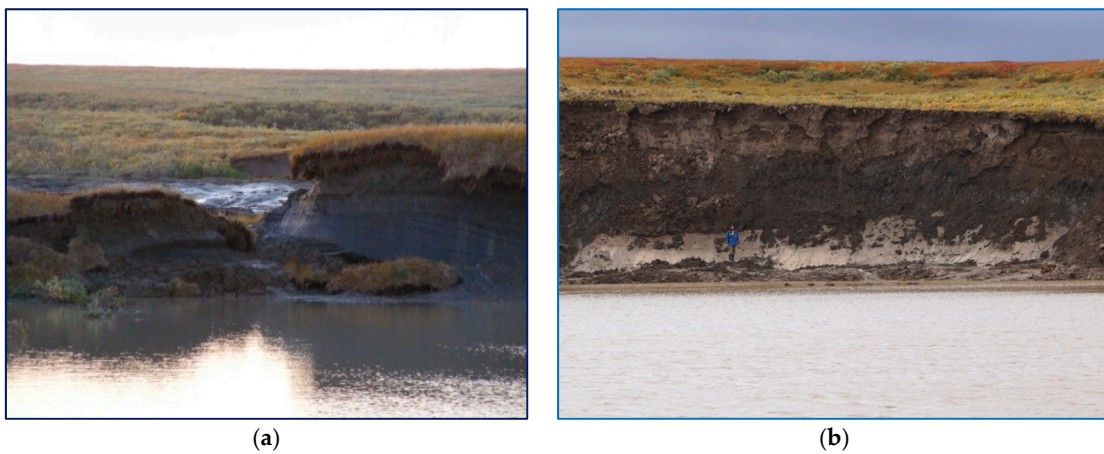

(**a**)    (**b**)

**Figure 2.** GEC lake walls with tabular ground ice: (**a**) GEC-1; (**b**) GEC-2.

The organic material that was abundant in the water and sediments of the GEC lakes originated from the shrub litter and mossy ground cover of the surface that was broken and ejected by the outburst of the mound-predecessor of the craters and then fell back into the crater cavity, as well as from dispersed organic matter of permafrost [2]. The organic matter and methane that were dissolved in the GEC lake water were measured during previous field studies [2]. The parameters of the GEC lakes and those of the background lakes are listed in Table 1.

While the rainwater and snow water were rather fresh, with a mineralization of around 5–10 $mg \cdot L^{-1}$, the river water, recharged by surface and subsurface runoff, had a mineralization exceeding 140 $mg \cdot L^{-1}$. This is especially clear in comparison to runoff from the thermocirque, in which saline marine deposits were exposed (Table 1). For this reason, both the river water and the lake water had a high mineralization compared to the atmospheric precipitation, and reached 470 $mg \cdot L^{-1}$, depending on the input from the coastal sediments caused by thermodenudation activity [12]. This input also controlled whether the continental ($Ca^{2+}$ and $HCO_3^-$ ions) signal or the marine (Na and Cl ions) signal was predominant in the lake water. The same signal was observed in those GEC lakes that had input from both marine (deeper layers of the section exposed at the early stages of GEC development) and continental deposits covering the geological section that was still exposed to the action of lake water at the later stages of GEC development.

The concentration of dissolved organic carbon (DOC) measured in GEC-1 decreased with time from 50.3 $mg \cdot L^{-1}$ in 2015, which was almost 10 times higher than that in other Yamal lakes, to 2–13 $mg \cdot L^{-1}$ in 2016 and 2017, which was still higher than the average value of 4.6 $mg \cdot L^{-1}$ for the background Yamal lakes (Table 1). Given that deposits from thermocirques can provide a rather high amount of DOC (243 $mg \cdot L^{-1}$, see Table 1), it is clear that, while the GEC was being inundated with the consequent preservation of saline marine deposits, the DOC concentration was decreasing.

**Table 1.** The geochemical parameters of the water in various water bodies of the Yamal peninsula. The range is summarized for the 2014–2017 sampling years.

| Water Object | Total Concentration of Main Ions, mg·L$^{-1}$ | Na+ mg/eq % | Cl$^-$ mg/eq % | Ca$^{2+}$ mg/eq % | HCO$_3^-$ mg/eq % | DOC mg·L$^{-1}$ | t °C Winter/Summer | pH | CH$_4$ μmol·L$^{-1}$ Winter/Summer |
|---|---|---|---|---|---|---|---|---|---|
| GEC lakes | 152–402 | 52–87 | 18–63 | 3–21 | 0–70 | 4.3–50.4 | (−0.08)/12.7 | 6.61–7.96 | 0.04–36.8/0.4–43.2 |
| GEC ice walls | 3–201 | 28–78 | 4–48 | 5–52 | 1–65 | 5.6–31.5 | - | - | 0.08–93.7 |
| Background lakes | 25–471 | 27–92 | 13–98 | 2–50 | 0–84 | 2.7–14.2 | 1.2/16.9 | 5.7–8.7 | 0.09–341/0–4.5 |
| Lake ice | - | - | - | - | - | - | - | - | 0.05–1.84 |
| Ponds | 1381–1946 | 47–71 | 87–94 | 11–23 | 0–4 | 7.6–10.9 | - | 5.9–7.43 | - |
| Rain | 5.2 | 8–31 | 41–50 | 45–55 | 0.2–0.4 | - | - | - | - |
| River | 141–147 | 55–58 | 56–91 | 14 | 6–42 | - | - | - | - |
| Snow | 4.4–51 | 12–71 | 24–86 | 7–43 | 3–56 | - | - | 5.19–7.52 | - |
| Thermocirque | 988 | 45 | 24 | 22 | 66 | 243 | - | 7.6 | - |

DOC: Dissolved organic carbon.

Due to the accepted hypothesis of methane's role in GEC formation [3], the methane concentration in the water of GEC lakes compared to that in the water of background lakes is their key geochemical feature. During 3–4 years after GEC formation, the methane source at the bottom was preserved in summer at the level of 43.2 $\mu$mol$\cdot$L$^{-1}$, while, in the background lakes, the highest value was only 4.5 $\mu$mol$\cdot$L$^{-1}$ (excluding shallow peat lakes and ponds with extremely high concentrations of both DOC and methane) (Table 1).

A comparison of the two GEC lakes (GEC-1 and GEC-2) with the two background lakes (LK001 and LK015) is presented in in Table 2.

**Table 2.** The parameters of the sampled lakes (LK001, LK015, GEC-1, and GEC-2) in April 2018.

| Parameter | LK001 | LK015 | GEC-1 | GEC-2 |
|---|---|---|---|---|
| Geology | IV[th] coastal-marine plain, clayey-silty deposits, altitude 40–41 m (Baltic) | | Concave slope within IV[th] coastal-marine plain, clayey-silty deposits, altitude 40–43 m (Baltic) | |
| lake surface's average altitude (Baltic), m | 12.8 | 11.4 | 33 | |
| lake area, ha | 37.16 | 9.92 | 0.57 | 1.35 |
| lake depth mean/max | 4.4/16.9 | 7.7/23.2 | 2.6/4.9 | 1.0/2.5 |
| Coastal lithology | Clay, silt, sand, tabular ground ice covered by talus | Clay, silt, peat, ice wedges, tabular ground ice | Clay, silt, tabular ground ice | |
| Leading coastal process | Thermoerosion | Thermodenudation | Coastal thermoerosion | |
| DOC, mg$\cdot$L$^{-1}$ (SF/BO) | 3.6/4.2 | 4.6/5.0 | 10.1/10.5 | 5.8 |
| Methane, $\mu$mol$\cdot$L$^{-1}$ (SF/BO) | 0.25/6.1 | 0.13/62.8 | 16/21 $\pm$ 0.2 | 0.42 |
| Water temperature | 1.4–2.0 | 1.3–3.1 $^\circ$C | 0.2 | 0.3 |

SF: surface water, BO: bottom water.

While the geomorphology of both lakes was rather similar (Table 2), the lake LK015 with an inflow of DOC from the thermocirque was smaller compared to lake LK001, and the impact from the surface runoff was noticeable: the DOC concentration was higher, and the methane concentration was much higher compared to the larger lake (LK001) with rather stable coasts (Table 2).

Both GEC lakes were located within concave slopes (erosion valleys) on the same geomorphic structure. Due to the 1-year time interval between the formation of GEC-2 (2012) and GEC-1 (2013), they differed in depth (the younger being deeper) and size (the younger being smaller). Probably, for this reason, GEC-1 had a higher DOC, and higher methane concentrations, than GEC-2.

Thus, the comparison included two background lakes with different sources and concentrations of soluble salts, organic matter, and methane, and two GEC lakes with different times of formation, and thus sizes, and concentrations of organic matter and methane.

## 3. Materials and Methods

### 3.1. Sample Collection and Characterization

Sampling was carried out in April 2018. Water and bottom sediments were collected through a hole drilled in the ice. Ice drilling was carried out in the deepest places of the lakes, according to bathymetric studies conducted earlier during summer surveys. Water was sampled using a TD-Automatika© hydrological water sampler, dispensed into 35-mL glass vials, sealed with gas-tight rubber stoppers (avoiding gas bubbles), and covered with a perforated aluminum cap. Bottom sediment (with a core length of up to 340 mm) were collected using a limnological stratometer with a glass tube. Sediment samples were then transferred (preserving the sediment's structure) into cut-off 5-mL plastic syringes and sealed with gas-tight rubber stoppers. Samples of water and sediments were stored in a portable temperature-controlled box at +20 $^\circ$C.

### 3.2. Analytical Techniques

The temperature and concentration of dissolved oxygen were measured with a WTW© 340i A HANNA HI8314F (Wensoket, RI, USA) portable ionometer with temperature compensation; a combination electrode was used for pH measurement. Specific conductivity was determined with a HANNA HI8733 (Wensoket, RI, USA) portable conductometer. Pore water was obtained by centrifugation of the sediments at 8000 g for 10 min. Methane content in the water and sediment samples was determined using the head-space method [19]. Methane concentration was measured on a Kristall-2000-M (Chromatec, Yoshkar-Ola, Russia) gas chromatograph equipped with a flame ionization detector. Three samples were used to obtain average values.

### 3.3. Isotopic Composition of Carbon Compounds

To determine the isotopic carbon composition of the suspended organic matter ($\delta^{13}$C-Corg), water samples were filtered through calcined 47 mm GF/F filters, which were then dried at 60 °C. The filtrate was used to determine the carbon isotopic composition of the dissolved bicarbonate ($\delta^{13}$C- $HCO_3{}^-$). The $\delta^{13}$C values were measured using a Delta Plus mass spectrometer (Thermo Electron Corporation, Langenselbold, Germany), using a PDB-calibrated standard. For methane, $\delta^{13}$C-$CH_4$ was measured on a TRACE GC gas chromatograph (Thermo Fisher Scientific, Waltham, MA, USA) coupled to a Delta Plus mass spectrometer. The error of $\delta^{13}$C measurements did not exceed $\pm 0.1$‰.

### 3.4. Radiotracer Experiments

The rates of microbial processes of sulfate reduction (SR), methanogenesis (MG), and methane oxidation (MO) were determined radioisotopically using labeled compounds: $NaH^{14}CO_3$, specific activity 2.04 GBq mmol$^{-1}$, Amersham, UK (10 µCi per sample), $^{14}CH_4$, specific activity 1.16 GBq mmol$^{-1}$, JSC Isotope, Russia (1 µCi per sample), and $Na_2{}^{35}SO_4$, specific activity 370 mBq mmol$^{-1}$, Perkin Elmer, USA (10 µCi per sample). A labeled substrate (0.2 mL as a sterile degassed water solution) was injected through the rubber stopper with a syringe. The vials were incubated for 20 h at in situ temperature (+4 °C). After incubation, the microbial processes (SR, MG, MO) were stopped by injecting 0.5 mL of saturated KOH solution into each experimental vial. All experiments were performed in duplicate.

After the end of the experiments, the vials were stored at 5–10 °C. Measurement of the radioactivity of the products of microbial activity in both the experimental and control vials was performed in the laboratory according to methods described earlier [20].

### 3.5. Cell Counts

The total microbial number (TMN) and cell size and shape were determined in the samples fixed with glutaraldehyde (2% final concentration). The fixed specimen (1–5 mL) was filtered through 0.2-µm black polycarbonate membranes (Millipore). The filters were stained with acridine orange and examined at ×1000 magnification under an Olympus BX 41 (Tokyo, Japan) epifluorescence microscope equipped with the Image Scope Color (M) visualization system.

### 3.6. DNA Extraction and Sequencing and Read-Centric Analysis

To collect the microbial biomass, the water sample (500 mL) was passed through filters with a pore diameter of 0.22 µm. The filters were homogenized by triturating with liquid nitrogen, and the preparation of metagenomic DNA was isolated by a method based on lysis of the cells followed by treatment with a detergent. A total of about 0.5 µg of DNA was isolated.

The oligonucleotide primers used for this experiment were 5′-<u>TCGTCGGCAGCGTCAGA TGTGTATAAGAGACAG</u> CCTACGGGNGGCWGCAG-3′ and 5′-<u>GTCTCGTGGGCTCGGAGAT GTGTATAAGAGACAG</u> GACTACHVGGGTATCTAATCC-3′, where the underlined regions are the Illumina adapter overhang nucleotide sequences, while the non-underlined sequences are

locus-specific sequences targeting conserved regions within the V3 and V4 domains of prokaryotic 16S rRNA genes. The locus-specific target sequences were designed based on a reported primer pair, namely S-D-Bact-0341-b-S-17 and S-D-Bact-0785-a-A-21 [21]. The amplified fragments were quantified with the Qubit dsDNA HS Assay Kit (Invitrogen, Merelbeke, Belgium) on a Qubit Fluorometer prior to sequencing. Paired-end sequencing of the library was performed on an Illumina MiSeq sequencer (San Diego, CA, USA) using the MiSeq Reagent Kit (v3) with the longest read length set to $2 \times 300$ base pairs (bp). A total of 407,923 reads were sequenced, with an overall length of 128,903,668 bp.

Before clustering, paired intersecting reads were combined into longer ones using the flash program. Low-quality readings and one-time readings (singletons) were removed from the analysis. Operational taxonomic unit (OTU) clustering was performed using the Usearch program for an identity level of 97%. To estimate the size of clusters, they were superimposed on the original combined reads, including singleton and low-quality readings, with a minimum identity of 97%. OTU taxonomic identification was performed by comparing them with the Silva 16S rRNA database [22], and searching for close sequences in GenBank using the BLASTN protocol. When a sequence with more than 95% similarity with the 16S rRNA gene of the described micro-organism was detected, OTU was assigned to the corresponding genus. A total of 234 OTU were classified, which included 168,729 sequences of 16S rRNA genes.

## 4. Results

Figure 3 presents the average concentrations of dissolved methane in the bottom water (a) and the bottom sediments (b) of all lakes studied: the background lakes (LK001, LK015) and the gas emission craters (GEC-1, GEC-2). The highest methane content in the bottom water was found in the lake LK015 ($9.1 \ \mu mol^{-1}$), and the lowest in the lake LK001 ($3.05 \ \mu mol^{-1}$). There were no differences between methane concentrations in the bottom water between background lakes and GEC lakes. In the upper sediment layer, the methane concentration was higher in both background lakes ($340/450 \ \mu mol^{-1}$) than in the GEC lakes ($45/62 \ \mu mol^{-1}$).

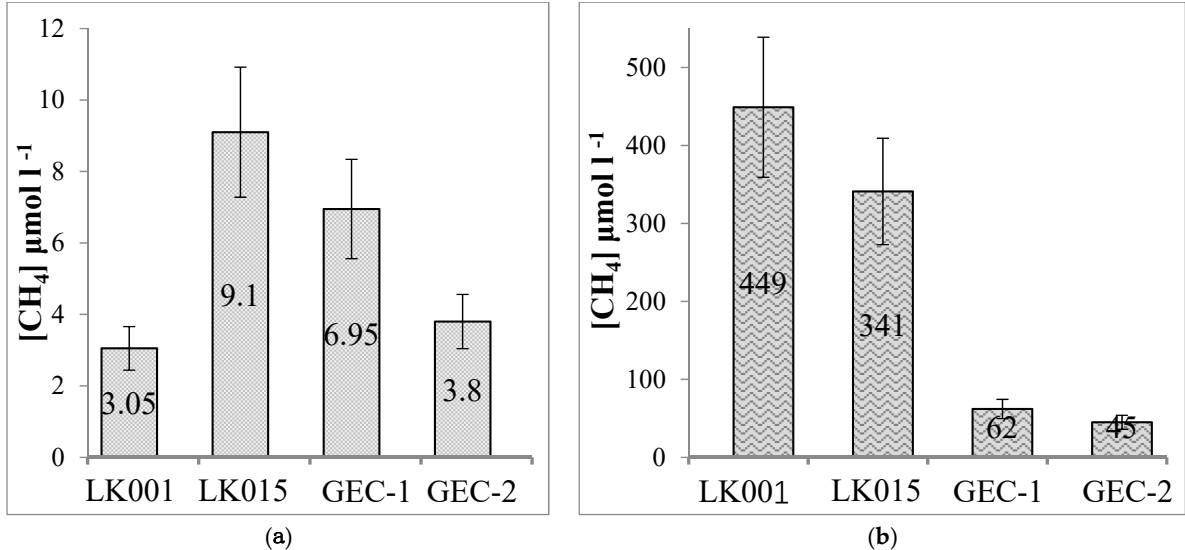

**Figure 3.** The average concentration of dissolved methane in the bottom water (**a**) and upper layer of the bottom sediments (**b**) of the Yamal background lakes (LK001, LK015) and gas emission crater lakes (GEC-1, GEC-2) (April 2018).

The isotopic composition of methane carbon in the bottom water and bottom sediments of all studied lakes was in a range from $-89.1‰$ to $-71.3‰$ (Figure 4). It should be noted that the methane carbon in the sediments of the background lakes was richer in the light isotope ($\delta^{13}C \ CH_4 = -89‰/-84‰$) than the methane carbon of the bottom water of all lakes ($\delta^{13}C \ CH_4 = -72‰/-71‰$).

In both background lakes, a reliable difference in the isotopic composition of the methane carbon in the sediments and bottom water was clearly visible ($\Delta\delta^{13}$C CH$_4$ = 13.1‰ and 17.0‰). For the GEC lakes, such a difference was less pronounced ($\Delta\delta^{13}$C CH$_4$ = 3.0‰ and 7.8‰).

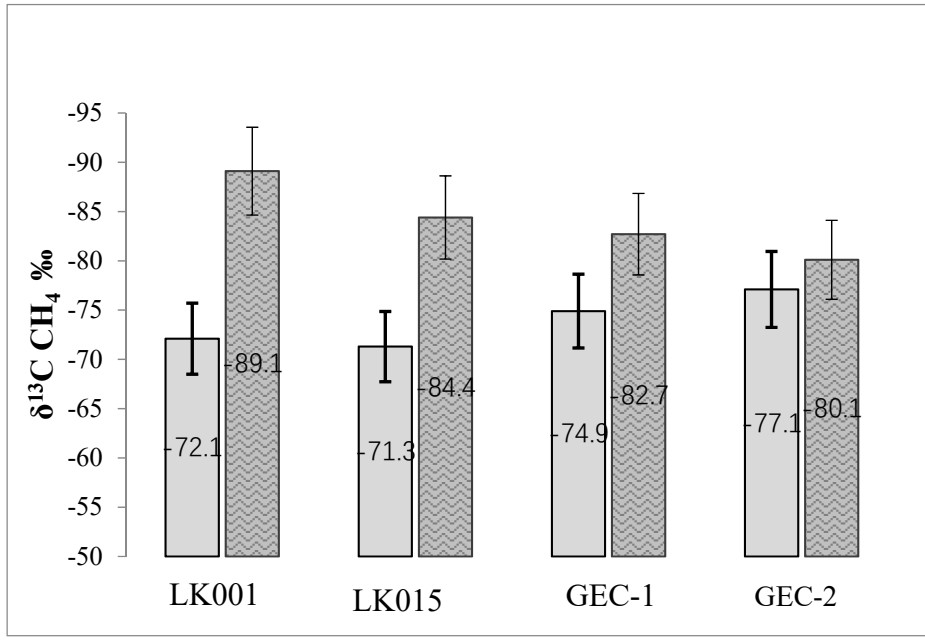

**Figure 4.** The average values of the isotopic composition of the methane carbon ($\delta^{13}$C CH$_4$ ‰) in the bottom water (light shadowing) and the bottom sediments from 0 to 14-cm depth (dark shadowing) of the Yamal background lakes (LK001, LK015) and gas emission crater lakes (GEC-1, GEC-2) (April 2018).

The rates of biogenic methane production via hydrogenotrophic methanogenesis (MG), methane oxidation (MO), and sulfate reduction (SR) were studied in the bottom sediment layer (0–14-cm depth) for all lakes by using a radiotracer technique (Figure 5). Although the bottom water and sediments were oxidized (Eh +40/+100 mV), activity of anaerobic processes of MG and SR were observed there (Figure 5a,c).

In the bottom sediments of the background lakes, the rates of MG were small (26–48 nmol·C·L$^{-1}$·day$^{-1}$) and corresponded to the known interval (50–400 nmol·C·L$^{-1}$·day$^{-1}$) for the winter season [23]. MG rates in the GEC sediments were very low and close to the resolution of the method (Figure 5a). The rate of MO in the bottom sediment layer of lake LK001 was similar (131/179 nmol l$^{-1}$ day$^{-1}$) to those found in both GEC lakes (Figure 5b). However, the MO rate in the lake LK015 was 3 times lower (average value 54 nmol·L$^{-1}$·day$^{-1}$). The SR rates ranged from 6 to 38 nmol·L$^{-1}$·day$^{-1}$ (Figure 5c), and were 3–6 times higher in the sediments of the background lakes compared with the GEC lakes.

The total number of micro-organisms (TNM) in the bottom water of the studied lakes was rather low (150–420 × 10$^3$ cell·ml$^{-1}$) (Table 3). Small cocci (0.4/0.5 μm) and short rods (0.4/0.8 μm) were predominant in all samples. High numbers of autofluorescent (AOF) cells were found in both background lakes, which were two times higher than those in the GEC lakes.

**Table 3.** The total number of micro-organisms (TNM) and number of autofluorescent cells (AOF) in the bottom water of Yamal lakes in April 2018.

| Lake | TNM (10$^3$ Cell·mL$^{-1}$) | AOF (10$^3$ Cell·mL$^{-1}$) |
|------|------|------|
| LK001 | 200 ± 50 | 60 ± 20 |
| LK015 | 340 ± 70 | 70 ± 20 |
| GEC-1 | 420 ± 70 | 30 ± 10 |
| GEC-2 | 150 ± 40 | 25 ± 10 |

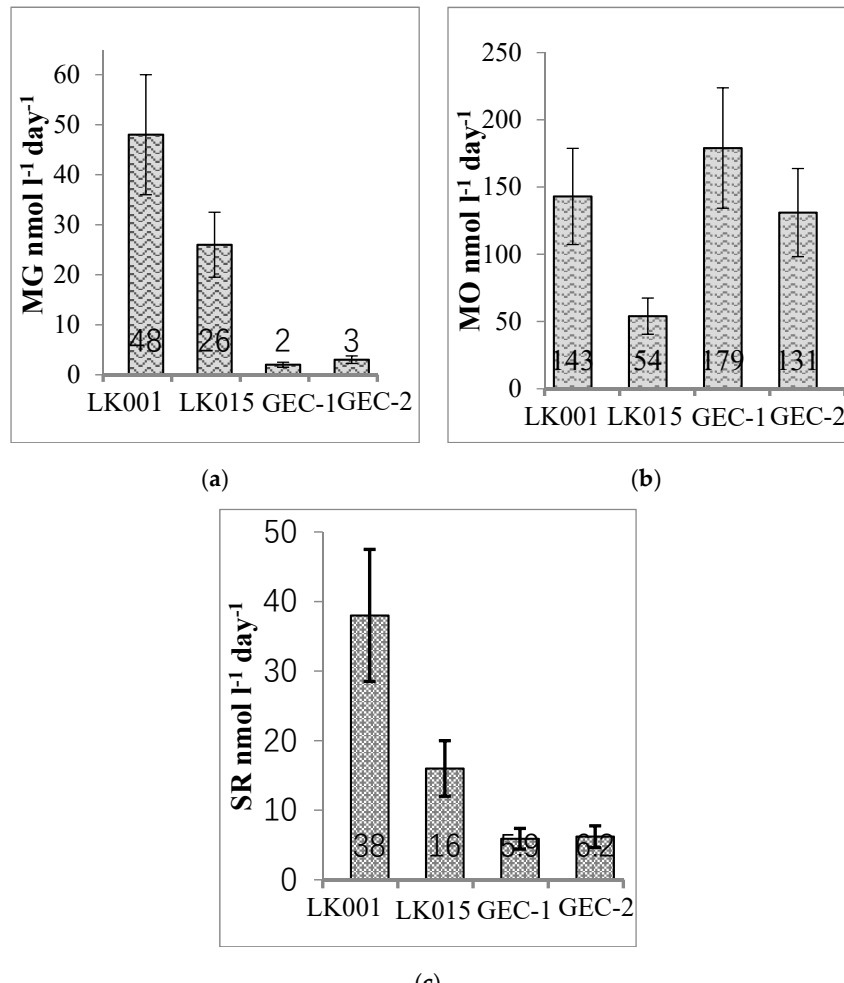

**Figure 5.** The average rates of the microbial processes of methane production (MG, (**a**)), methane oxidation (MO, (**b**)), and sulfate reduction (SR, (**c**)) in the bottom sediments of the Yamal background lakes (LK001, LK015) and gas emission crater lakes (GEC-1, GEC-2) in April 2018.

A total of 168,729 sequences of the 16S rRNA gene fragments from the bottom water and the upper layer of the sediment of background lakes and GEC lakes were obtained. An analysis of the 16S rRNA data revealed high taxonomic diversity of microbial communities in all studied lakes (Figure 6).

The microbial communities of the bottom water of the background and GEC lakes were quite similar (Figure 6a). A large proportion of micro-organisms belonged to bacteria (99.97–99.06% to total 16S rRNA reads). Among those, the highest relative abundance was attributed to *Actinobacteria* (30.42–41.4%), *Bacteroidetes* (4.62–17.73%), *Betaproteobacteria* (11.37–23.99%), and *Gammaproteobacteria* (15.23–47.53%). *Acidobacteria* accounted for more than 3% only in LK015 water; all other bacterial taxa accounted for less than 3%. Members of the phyla *Firmicutes, Gemmatimonadetes, Nitrospirae, Epsilonproteobacteria, Chloroflexi*, and SR1 composed a minor part of the water microbial communities with the contribution of each taxon of <1% (referred to as 'other bacteria' in Figure 6). The residual archaea were mainly represented by uncultivable forms.

In contrast to water, archaea had the highest relative abundance in the sediments of both background lakes (50–59% of the total 16S rRNA reads). In the GEC sediments, archaea represented only up to 11%; however, they still were more numerous than in bottom water. Other taxa abundant in the sediments were *Bacteroidetes* (16.6–26.3%), *Betaproteobacteria* (1.3–21%), *Acidobacteria* (0.9–17.8%), and *Actinobacteria* (3.9–5% in GEC sediments only). Representatives of the phyla *Firmicutes, Alphaproteobacteria, Verrucomicrobia, Chloroflexi*, and *Planctomycetes* accounted for less than 1% (Figure 6b).

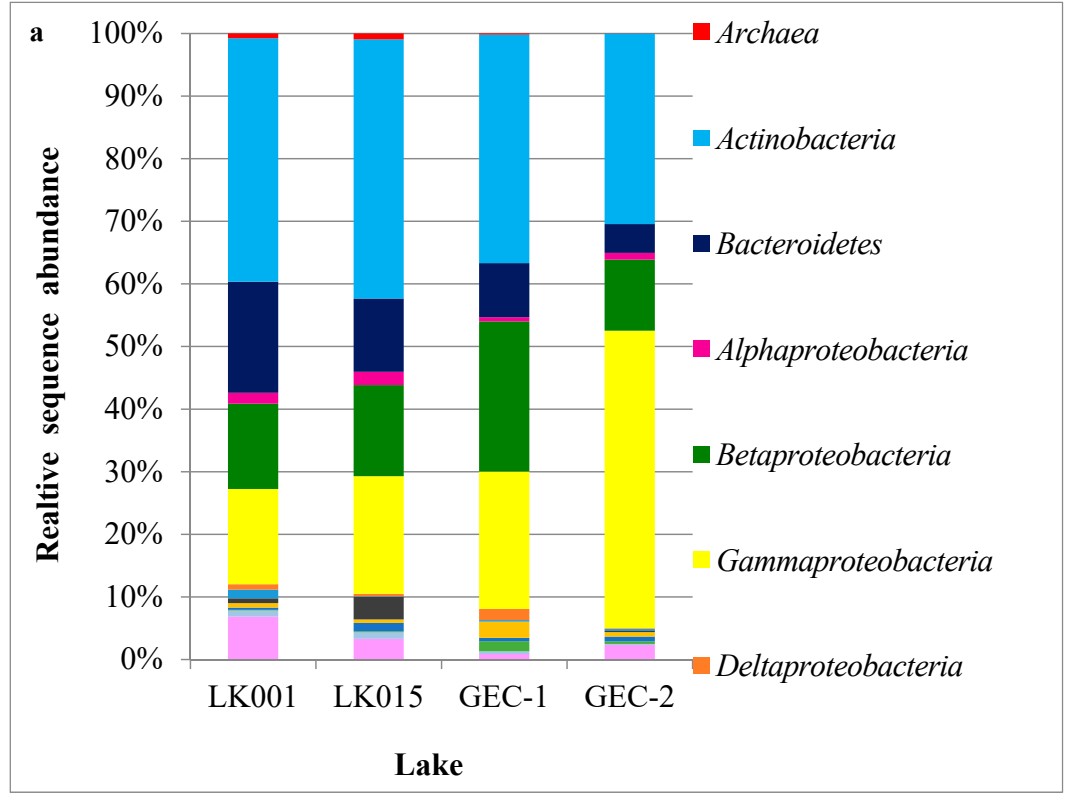

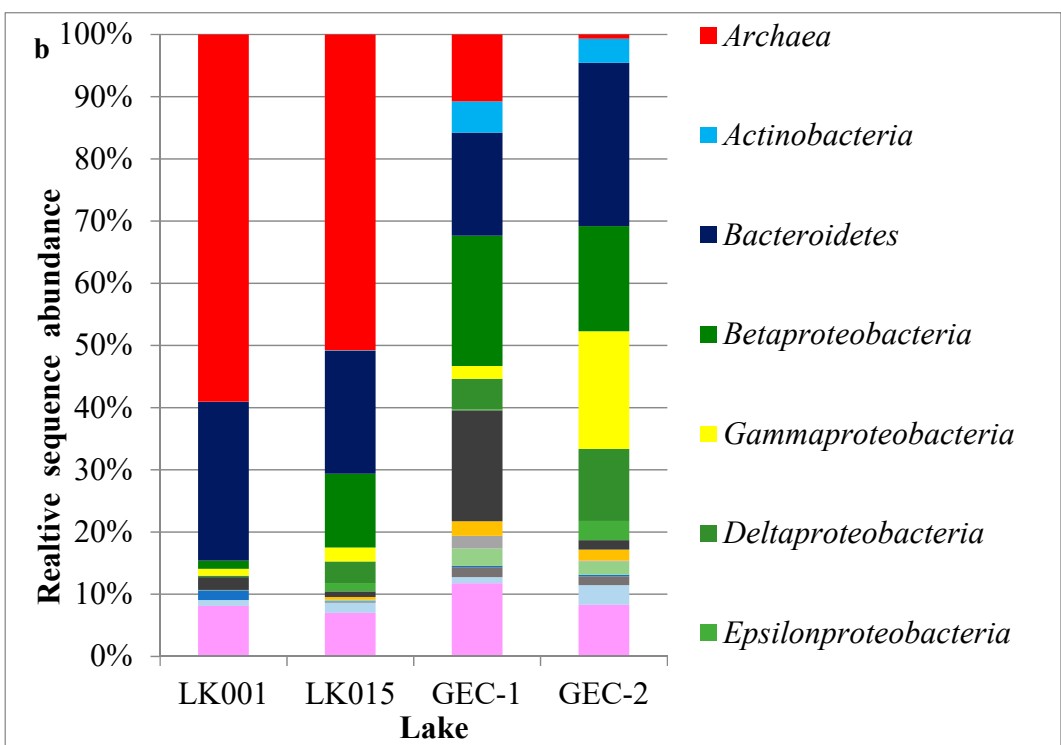

**Figure 6.** Microbial diversity in the bottom water (**a**) and upper sediment layer (**b**) of the Yamal background lakes (LK001, LK015) and gas emission craters (GEC-1, GEC-2) as determined by high-throughput sequencing of the 16S rRNA gene fragments (April 2018).

Table 4 presents the relative abundances of bacteria affiliated with previously described genera with known physiological functions in the microbial community. For example, among the dominant

bacterial taxa, *Actinobacteria* were mainly represented by planktonic aerobic chemoorganotrophic bacteria of the order *Candidatus* "Nanopelagicales", which is found in various aquatic habitats [24]. Their contribution in the sediments was very low except for in the GEC lakes, where Ca. "Planktophila" accounted for 3–5%.

**Table 4.** The physiological affiliation of major bacterial phyla detected in Yamal lakes.

| Sample | Bottom Water | | | | Bottom Sediments (0–14-cm Depth) | | | |
|---|---|---|---|---|---|---|---|---|
| Lake | LK001 | LK015 | GEC-1 | GEC-2 | LK001 | LK015 | GEC-1 | GEC-2 |
| Chemoorganotrophic *Actinobacteria* (% to total 16S rRNA reads) | | | | | | | | |
| *Ilumatobacter* | 9.1 | 8.5 | 15.5 | 11.9 | 0 | 0 | 0.1 | 0.6 |
| Ca. "Nanopelagicus" | 10.4 | 7.9 | 0.06 | 0 | 0.01 | 0 | 0 | 0 |
| Ca. *"Planktophila"* | 11.9 | 17.7 | 16.8 | 13.4 | 0.01 | 0.07 | 4.6 | 3.2 |
| *Betaproteobacteria* (% to total 16S rRNA reads) | | | | | | | | |
| Methylotrophic | 3.5 | 8.1 | 1.7 | 1.4 | 0.16 | 0.16 | 0.2 | 0.7 |
| Involved in N cycle | 5.7 | 1.7 | 0.5 | 1.2 | 0.17 | 0.3 | 4.8 | 0.6 |
| Ferrous-iron-oxidizing | 2.0 | 0.7 | 0 | 0.01 | 0.08 | 3.9 | 9.98 | 1.1 |
| Chemoorganotrophic | 2.36 | 3.95 | 20.97 | 8.7 | 0.25 | 4.5 | 5.9 | 13.2 |
| Sulfur-oxidizing | 0 | 0 | 0.01 | 0 | 0.7 | 3.0 | 0.04 | 0.07 |
| *Gammaproteobacteria* (% to total 16S rRNA reads) | | | | | | | | |
| Methanotrophic | 7.6 | 18.0 | 17.9 | 9.1 | 1.04 | 0.3 | 1.8 | 1.05 |
| Chemoorganotrophic | 7.0 | 0.03 | 3.5 | 38.2 | 0.01 | 1.9 | 0.1 | 17.8 |
| *Deltaproteobacteria* (% to total 16S rRNA reads) | | | | | | | | |
| Sulfate-reducing | 0 | 0 | 0 | 0 | 0.11 | 2.53 | 2.2 | 4.3 |

Some physiological groups were poorly represented (especially in sediments) due to the low abundance of micro-organisms closely related to those with known physiology in our samples. In contrast, the taxa related to unculturable micro-organisms with yet unknown physiological properties were not included in Table 4. For example, most of the sequences assigned to *Bacteroidetes* were represented by various "uncultivable" lines phylogenetically distant from known groups. Cultivated representatives belonging to aerobic chemoorganotrophic bacteria of the genera *Flavobacterium* and *Sediminibacterium* were also found.

Among the *Proteobacteria* phylum, two classes were dominant: *Beta-* and *Gammaproteobacteria* (Table 4). The share of methylotrophic bacteria of the genera *Methylophilus*, *Methylotenera*, and Ca. "Matholloputilos" (class *Betaproteobacteria*) in the bottom water of the background lakes was higher than that in the GEC lakes. These bacteria use $C_1$-methylated compounds as their growth substrates. *Betaproteobacteria* involved in various stages of the nitrogen cycle and aerobic iron oxidation (*Nitrotoga*, *Nitrospira*, *Gallionella*) were also detected. On the other hand, chemoorganotrophic betaproteobacteria related to those detected in various cold habitats (*Limnohabitans*, *Polaromonas*) prevailed in the bottom water of the GEC lakes. Chemoorganotrophic and ferrous-iron-oxidizing betaproteobacteria were also numerous in the sediments of the studied lakes.

A large relative amount of aerobic methanotrophic *Gammaproteobacteria* was detected in all samples of the bottom water (Table 4). They were closely related to psychrotolerant *Methylobacter tundripaludum* isolated from Norwegian Arctic swamp soils [25]. Aerobic methanotrophs were also detected in the sediments, where they comprised up to 1.8% of the total 16S rRNA reads. In sediments, the MO rates and relative abundances of aerobic methanotrophs correlated well with each other. It can be seen from a comparison of the calculated ratios between the actual and lowest values for the MO rates, which were 2.65 (LK001), 1 (LK015), 3.3 (GEC-1), 2.4. (GEC-2), and for methanotroph abundances 3.5 (LK001), 1 (LK015), 6 (GEC-1), 3.5 (GEC-2). It is important to stress that relative abundances provide no information about the physiological state of the cells. Some cells from which DNA was extracted could be inactive, dormant, or dead.

Heterotrophic gammaproteobacteria, mainly represented by free-living pseudomonads, were abundant in the bottom water of lakes LK001, GEC-1, and GEC-2, as well as in the sediments of GEC-2. Sulfate-reducing bacteria were poorly represented: Only *Desulfuromonas* sp. in low relative abundance.

Methanogenic archaea were the major contributors to the sediment microbial communities in both background lakes and GEC-1, and their relative abundance was much higher than in the bottom water. The share of methanogens was about one half of all archaea in the sediments (Table 5).

**Table 5.** The major representatives of the phylum *Archaea* in the microbial community of Yamal lakes.

| Sample | Near Bottom Water | | | | Bottom Sediments (0–14-cm Depth) | | | |
|---|---|---|---|---|---|---|---|---|
| Lake | LK001 | LK015 | GEC-1 | GEC-2 | LK001 | LK015 | GEC-1 | GEC-2 |
| % to Total 16S rRNA Reads | | | | | | | | |
| *Methanoregula* | 0.12 | 0.16 | 0.05 | 0.02 | 22.93 | 15.27 | 0.06 | 0.08 |
| *Methanosarcina* | 0 | 0 | 0.01 | 0 | 0.04 | 0.78 | 2.91 | 0.13 |
| *Methanosaeta* | 0 | 0 | 0 | 0 | 3.61 | 5.69 | 0.64 | 0.01 |
| *Methanomassiliicoccus* | 0 | 0 | 0 | 0 | 1.18 | 0.88 | 0.06 | 0 |
| ANME-2d | 0 | 0 | 0 | 0.15 | 1.80 | 0.04 | 2.27 | 0.15 |
| Other archaea | 0.63 | 0.77 | 0.12 | 0.29 | 29.49 | 28.14 | 4.83 | 0.29 |
| Total archaea | 0.76 | 0.94 | 0.18 | 0.67 | 59.05 | 50.80 | 10.77 | 0.67 |

The share of archaea in the sediments of the background lakes was 51–59%, while in the sediments of the GEC lakes it was only 1–10%. This is the most powerful indicator of the differences in the composition of microbial communities in the background and GEC lakes. Most archaea were represented by hydrogenotrophic methanogens of the genus *Methanoregula* and acetoclastic methanogens of the genera *Methanosarcina* and *Methanotrix* (*Methanosaeta)*. The remaining archaea were unculturable species with unknown metabolism.

## 5. Discussion

Biogeochemical, microbiological, and molecular diagnostic studies were undertaken in four water bodies of the Yamal Peninsula, two lakes inside gas emission craters and two tundra lakes, chosen for the control, relatively close in size to the GEC lakes. All of the lakes did not substantially differ in such hydrological characteristics as temperature, mineralization, pH, and oxygen concentration. The dissolved methane concentration in the bottom water of all four lakes studied varied within a rather narrow range (Figure 3a).

Yet, a number of parameters showed substantial differences between the GEC lakes and the background lakes. Methane concentrations in the bottom sediments were within the wide range of concentrations known for mesotrophic and dystrophic lakes of the boreal zone [26]. At the same time, the methane concentration in the background lakes was reliably higher than that in the bottom sediments of the GEC lakes (Figure 3b). Such a difference is related to the fact that, in the bottom sediments of background lakes, a native community of micro-organisms was formed, which included methanogenic archaea. Organic matter settled from the water column was sufficient for the functioning of anaerobic processes, which probably are occurring within the oxygen-free microzones, although, in general, the conditions in the bottom sediments were weakly oxidized. In the young sediments of the GEC lakes, the conditions sufficient for the existence of a methanogenic community have not been satisfied yet, which was proven by low MG rates and a low methane concentration (Figure 5a). Inoculation of the sediments by micro-organisms from surrounding soil has just begun. A small amount of archaea found in the GEC lake sediments were probably allochtonous microbiota from the surface and active-layer runoff. Permafrost thaw leads to a great transfer of soil microbes into aquatic communities [27,28].

A quarter of all archaea in the sediments of background lakes belonged to hydrogenotrophic methanogens of the genus *Methanoregula* and acetoclastic *Methanosarcina* and *Methanosaeta*; both groups have been commonly detected in various thermokarst lakes [29–31]. In general, all four known pathways

for methane production (hydrogenotrophic, aceticlastic, methylotrophic, and methyl-reducing) were previously reported for permafrost-related environments, including lakes [28–33]. The low concentration of the heavy isotope in methane carbon in both bottom water and sediments (−71‰/−89‰) indicated the definitely biogenic origin of this gas [34,35]. Yet, these data do not provide for determination of the age of the methane; specifically, whether it was formed before the permafrost's development or is produced now within the sediments of the studied lakes.

Among the common features for all studied lakes was a difference in the isotope composition of methane carbon in the bottom water and sediments (Figure 4). The concentration of the light isotope $^{12}$C was noticeably lower in the bottom water's methane carbon compared to the methane carbon of the sediments ($\Delta\delta^{13}$C = 13.1/17‰). Most likely, this results from isotope fractionation caused by preferential consumption of the light isotope of methane carbon in the bottom water by methanotrophic micro-organisms. Residual methane, therefore, became enriched in the heavy carbon isotope. The difference in the carbon isotope composition of dissolved methane between the sediments and the bottom water layer is the geochemical outcome of microbial processes (methane oxidation) over a relatively long period of time. However, this indicator does not necessarily correlate with the MO rate. The rate of MO is determined in experiments with a relatively short-term incubation of water and sediment samples. The effect of microbial fractionation of methane carbon isotopes was mild at the water–sediment interface of the lakes GEC-1 and GEC-2 (Figure 4), reflecting low activity of the methanotrophic community. The molecular genetic study showed methanotrophic gammaproteobacteria to be characteristic of the bottom water of all studied lakes. At the same time, in the sediments, their contribution was much lower than in the bottom water (Table 4). Yet, it does not mean that the MO process was mainly restricted to the water layer, because the actual number of micro-organisms in the sediments was substantially higher than in the bottom water. The highest MO rates are probably specifically associated with the water–sediment interface.

Microbial methane oxidation is a key process in methanogenic habitats, mitigating methane emission into the atmosphere. In the permafrost-associated environments, methanotrophic micro-organisms could be responsible for oxidation of 20–60% of the methane [36]. Aerobic methanotrophic *Gammaproteobacteria* in our samples were represented by psychrotolerant *Methylobacter tundripaludum*. Aerobic methanotrophic members of *Alpha-* (type II) and *Gammaproteobacteria* (type I) are typical inhabitants of freshwater lakes, and gammaproteobacterial *Methylobacter* spp. are the most frequently detected methanotrophs in various freshwater habitats, including boreal lakes [37,38]. Members of the *Methylococcaceae* family were also shown to be predominant in some thermokarst lakes of Alaska [29,39] and Canada [40]. Apart from aerobic methanotrophs, we detected various methylotrophic bacteria in the bottom water of the background lakes. Methylotrophic bacteria have often been shown to accompany methanotrophs in different environments, including thermokarst lakes [32,41].

Anaerobic oxidation of methane (AOM), probably by a representative of nitrate-dependent ANME-2d, was found in the sediments of all studied lakes. Despite the efforts of various scientists, AOM in thermokarst lakes is still a mysterious process. Some AOM representatives were detected in Alaskan lakes, albeit at a very low relative abundance. For example, nitrate-dependent archaea Ca. 'Methanoperedens' (ANME-2d) were present in the original sediment samples (lakes Emaiksoun and Unnamed, Alaska), indicating that AOM might be possible, while anaerobic methanotrophic activity with $NO_3-$ or $NO_2-$ as the electron acceptor was not detected. Together with low in situ $NO_3-$ concentration, this indicated that $NO_3-$ did not play an important role in this ecosystem [32]. ANME-2d together with sulfate-dependent ANME-2 a/b were also detected in another Alaskan Lake Sukok [29]. The most intriguing results were obtained for Alaskan Lake Vault, where aerobic methanotrophs of the genus *Methylobacter* were shown to be involved in AOM. As components of microbial communities, they oxidize under anoxic conditions up to 32% of the methane formed in the upper sediments of this shallow ferruginous lake [39].

One can note a substantial difference in the absolute values of MG and MO rates in the GEC lakes (Figure 5a,b). An obvious excess of oxidized methane over its production by MG requires an

explanation for the appearance of extra methane. Possible explanations can be either diffusion from the deeper layers of sediments beneath the sampled ones (deeper than 35 cm), or from the frozen deposits in the GEC walls.

It is known that, in lake sediments under dissolved oxygen limitation, methane formation and sulfate reduction occur simultaneously. The total carbon cycle and the microbial cycle of methane in particular are closely related to the microbial cycle of sulfur. The process of SR is observed even in those lake sediments with an extremely low concentration of sulfate ion [42]. Therefore, obtaining quantitative data on the intensity of SR in the sediments provides additional information on the overall activity of microbial processes. The quantitative values for the SR rates that we obtained turned out to be very low, even in comparison to similar data on the sediments of lake Baikal [42] and Gek-Gel lake in the Caucasus Mountains [43]. However, in this respect, the GEC lake sediments with extremely low SR rates differed from the sediments of the background lakes. Thus, one more parameter indicates the lack (underdevelopment) of the normal microbial community in the sediments of GEC lakes.

The rest of the bacterial representatives of the studied background and GEC lakes were mostly *Actinobacteria*, *Bacteroidetes*, *Betaproteobacteria*, and *Acidobacteria*, which is typical of other thermokarst lakes. The relative predominance of carbon-degrading *Bacteroidetes* was previously shown for a number of lakes [32,44]. *Betaproteobacteria* and *Actinobacteria* are also abundant taxa [29,44]. *Betaproteobacteria* and *Bacteriodetes* in aquatic environments are associated with organic-rich substrates and prefer labile carbon [45]. *Acidobacteria*, *Firmicutes*, *Verrumicrobia*, *Gemmatimonadetes*, and *Chlroflexi* were also detected in small Canadian thermokarst ponds [44]. A high abundance of the *Nitrosomonadales* ammonia-oxidizing bacteria and members of the order *Anaerolineales* were found in Alaskian lakes [32]. Methane-cycling (methanogens and methanotrophs) and N-cycling (*Nitrospirae*) organisms developed in a lake with methane of biogenic origin, whereas, in a lake with methane of thermogenic origin, the taxa typically involved in biogeochemical sulfur transformations and photosynthesis were dominant [29].

In contrast to the data on the microbial activity of the carbon and sulfur cycles, as well as the results of the analysis of the composition of microbial communities, the data on the total number of micro-organisms in the bottom water of both background and GEC lakes turned out to be similar. The total microbial numbers in all four lakes were close to that in the water column of oligotrophic lakes during the winter season [46].

Our study was undertaken in early spring, with the samples taken from under the lake ice, and characterizes only cold water processes. Summer sampling planned in the future may result in new findings concerning the intensity of microbial processes.

## 6. Conclusions

To summarize the novel data obtained in the first microbiological study of Central Yamal lakes, in comparison to specific lakes formed recently within the gas emission craters, we may state the following.

- Yamal lakes share similarities with the tundra and boreal lakes located in other areas of the permafrost zone. Slow-flowing microbial processes are characteristic of both types of lakes (GEC and background), which is expected for the cold climate of the studied area. The bacterial communities of both types of studied lakes were dominated by the taxa typically found in thermokarst lakes and other permafrost-affected habitats. Both types of studied lakes were inhabited by aerobic methanotrophs of the genus *Methylobacter*: the most frequently detected methanotrophs in various freshwater environments, including boreal thermokarst and non-thermokarst lakes. The nitrate-dependent ANME 2d detected in both GEC and background lakes were also found previously in other thermokarst lakes. Methane concentrations in the sediments of background lakes were within the wide range of concentrations known for mesotrophic and dystrophic lakes of the boreal zone, and representatives of methanogenic archaea in background lakes were similar to those found in other boreal basins.

- At the same time, the GEC lakes essentially differed from the background ones by low rates of anaerobic processes (methanogenesis and sulfate reduction), a reliably lower methane concentration, and low diversity and abundance of methanogenic archaea in the sediments. Archaea in the GEC lakes were probably allochthonous microbiota from the surface and active-layer runoff. Thus, GEC lakes could be distinguished from other exogenous lakes based on their weak methanogenic population and activity.

- It can be gingerly assumed that the very slow rates of anaerobic microbial processes indicate a transformation of the newly formed water bodies (i.e., GEC lakes) into real lakes. This may relate not only to GEC lakes, but also to newly formed thermokarst lakes.

**Author Contributions:** Conceptualization, A.S. and M.L.; Methodology, A.S. and M.L.; Field Investigation A.S., V.K., Y.D. and A.K. (Artem Khomutov); Laboratory Investigation A.S. and V.K., Resources, N.P.; Writing—Original Draft A.S., M.L. and A.K. (Anna Kallistova); Writing—Review & Editing A.S., M.L. and A.K. (Anna Kallistova); Supervision, N.P. and N.R.; Project Administration, N.P. and M.L.

**Funding:** Research on the composition of microbial communities and the rates of microbial processes was funded by the Russian Science Foundation, grant number 16-14-10201. General lake and crater information was obtained within the project funded by the Russian Science Foundation, grant number 16-17-10203.

**Acknowledgments:** The authors thank the "Russian Center for Arctic Development" for organizing and supporting the field work.

**Conflicts of Interest:** The authors declare no conflicts of interest.

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
