# Peer review of "Microbiological Study of Yamal Lakes: A Key to Understanding the Evolution of Gas Emission Craters"

_geosciences, doi:10.3390/geosciences8120478_

Round 1
Reviewer 1 Report
From my point of view, this manuscript is written with engough details with good English that can be considered to be published in its present form.
Author Response
We thank the Reviewer for the appreciation of our manuscript.
Reviewer 2 Report
The article by Savvichev and co-authors is devoted to an interesting phenomenon discovered in the permafrost. The authors attempted to find microbiological indicators that allow to determine the differences between the formations of thermokarst lakes and GEC lakes. The authors obtained original results, but the manuscript should be revised to a large extent.
First, the title of manuscript does not reflect the results and conclusions.
In the introduction, which should be reduced, it is necessary to clarify what is GEC in permafrost, how it differs from ordinary volcanic craters, whether it occurs everywhere in the permafrost zone, for example, in the Canadian Arctic, or only in certain areas of the Russian Arctic. And, finally, I strongly advise to clarify why it is important to know the differences in the origin of small Arctic lakes.
The description of the study area must be reduced. My recommendation to the authors is to either remove Table 1 from the text or make a major revision to it. This table raises a large number of questions: what do the asterisks mean, the dimension for oxygen concentration, to which do the gaps correspond - no data or 0, etc. In my opinion, tables 2 and 3 should be combined in order to really compare the characteristics of all objects.
From the Materials and Methods section it is unclear where exactly water and sediment samples were taken in each lake. Do samples of water and sludge represent averaged mixes for each lake? How were the samples stored before the research?
Table 6 should be expanded in terms of methanogens and the groups that the authors mention in the discussion. Namely, the presence or absence of methanogens is considered by the authors to be an indicator of the origin of the lakes, while the composition of bacteria changed little. However, table 5 with the detected bacterial physiological groups is very detailed as compared with the scant archaea data (table 6).
On the one hand, discussion should be significantly reduced; on the other hand, data on the detection of methane in the permafrost, in which these crater lakes were formed, should be brought to the discussion. According to the literary sources used by the authors, methane and methanogens in the permafrost began to be studied in the 2017-2018 years, although there are earlier works:
Rivkina E. et al. Biogeochemistry of methane and methanogenic archaea in permafrost. FEMS Microbial Ecology. 2007. V. 61. №1. P. 1-15.
Rivkina E.M., Samarkin V.A., Gilichinskiy D.A. Methane and permafrost soil of the Kolyma-
Indigir lowland. Eurasian Soil Science. 1992. V. 25. №1. P. 50-53.
Kraev G. N., Schultze E. D., Rivkina E. M. Cryogenesis as a factor of methane distribution in layers of permafrost. Doklady Earth Sciences. Springer US, 2013. Т. 451. №. 2. С. 882-885.
Rivkina E., Petrovskaya L., Vishnivetskaya T. et al. Metagenomic analyses of the late Pleistocene permafrost - Additional tools for reconstruction of environmental conditions. Biogeosciences. 2016. V. 13. №7. -P. 2207-2219.
Other minor suggestions:
gas emission crater should be written everywhere instead of gas-emission crater;
It should be everywhere written mg l-1 instead of mg /l;
In different parts of the manuscript, the authors designate the objects to be studied: GEC1 and GEC2 or GEC-1 and GEC-2. It is necessary to choose and label the same everywhere;
Page 2, line 51 and 54 change the references with the full list of authors;
Line 160 “methanogenesis H2/CO2” change on “methanogenesis”
Line 162 14CH4 …. mmol-1 change on 14CH4…..mmol-1
Line 165-166 What temperature was relevant?
Line 210-211The different and strange units for methane concentrations are given – μM and μmol-1?
Line 215 What is “average methane concentration”? How many samples did the authors use? The same question on line 223
Line 240 and line 291 C-1- methylated change on C1-methylated
Author Response
We thank the reviewer for valuable comments and advices on improving the manuscript. Please see the answers in the attached files.

Reviewer 3 Report
Dear editor, dear authors,
I have added my comments directly into the pdf-file.

Author Response
We thank the reviewer for valuable comments and suggestions for improving the text of the manuscript. Please see the answers in the attached file.

Round 2
Reviewer 3 Report
Dear authors,
I have re-read your revised manuscript and to my understanding it has significantly improved.
There are only two minor comments left, you explained to me my remarks/questions in the ms, however in two cases I think this explanation should also / better be incorporated into the ms-text.
Comment / response to table 4/5
MO rates in sediments and aerobic methanotrophs (MT) match well with each other. You can see it when compare the ratios: MO rate in sediments (143, 54, 179, 131 nmol/l/day), MT abundances in sediments (1.04, 0.3, 1.8, 1.05%), ratios for MO rate (2.65, 1, 3.3, 2.4), MT (3.5, 1, 6, 3.5). Here it is important to understand that relative abundance do say nothing about the physiological state of the cell. Some cells from which DNA was extracted could be inactive or dead.
Please clarify this aspect also in the manuscript text.
Page 13 of cover letter
But only in background lakes there was a substantial difference between bottom water and sediment. in GEC lakes the isotopic signal was similar.
But as MO-rates were similar in all lakes (expcept L15) the isotopic signal can not be explained by methan oxidation.
The difference in the carbon isotope composition of dissolved methane between sediments and the bottom water layer is the geochemical outcome of microbial processes (methane oxidation) over a relatively long period of time. However, this indicator is not obliged to correlate with the rate of the MO process. The rate of MO is determined in experiments with relatively short-term incubations of water and sediment samples.
Please clarify this aspect also in the manuscript text.
Author Response
Thank you very much for your attentive and critical reading of our manuscript.
We added into the text explanations according to your suggestion: P12, L324-329 (to table 4) and P14, L391-398 (to carbon isotope composition of dissolved methane), latter led to slight changes in the surrounding text.
Our Institute translator proofread English language.